# Graph Reinforcement Learning for Network Control via Bi-Level Optimization

**Daniele Gammelli**
Stanford University
gammelli@stanford.edu

**James Harrison**
Google Research, Brain Team
jamesharrison@google.com

**Kaidi Yang**
National University of Singapore
kaidi.yang@nus.edu.sg

**Marco Pavone**
Stanford University
pavone@stanford.edu

**Filipe Rodrigues**
Technical University of Denmark
rodr@dtu.dk

**Francisco C. Pereira**
Technical University of Denmark
camara@dtu.dk

## Abstract

Dynamic network flow models have been extensively studied and widely used in the past decades to formulate many problems with great real-world impact, such as transportation, supply chain management, power grid control, and more. Within this context, time-expansion techniques currently represent a generic approach for solving control problems over dynamic networks. However, the complexity of these methods does not allow traditional approaches to scale to large networks, especially when these need to be solved recursively over a receding horizon (e.g., to yield a sequence of actions in model predictive control). Moreover, tractable optimization-based approaches are limited to simple linear deterministic settings, and are not able to handle environments with stochastic, non-linear, or unknown dynamics. In this work, we present dynamic network flow problems through the lens of reinforcement learning and propose a graph network-based framework that can handle a wide variety of problems and learn efficient algorithms without significantly compromising optimality. Instead of a naive and poorly-scalable formulation, in which agent actions (and thus network outputs) consist of actions on edges, we present a two-phase decomposition. The first phase consists of an RL agent specifying *desired outcomes* to the actions. The second phase exploits the problem structure to solve a convex optimization problem and achieve (as best as possible) these desired outcomes. This formulation leads to dramatically improved scalability and performance. We further highlight a collection of features that are potentially desirable to system designers, investigate design decisions, and present experiments showing the utility, scalability, and flexibility of our framework.

## 1 Introduction

Many economically critical real-world systems are well-modelled through the lens of control on graphs. Power generation [1–3]; road, rail, and air transportation systems [4, 5]; complex manufacturing systems, supply chain, and distribution networks [6, 7]; telecommunication networks [8–10]; and many other systems are fundamentally the problem of controlling flows of products, vehicles, or other quantities on graph-structured networks. Traditionally, these problems are approached through the definition of a *dynamic network flow model* (DNF) [11, 12]. Within this class of problems, Ford and Fulkerson [13, 14] proposed a generic approach, showing how one can use time-expansion techniques to (i) convert dynamic networks with discrete time horizon into static networks, and (ii) solve the problem using algorithms developed for static networks. However, this approach leads to networks that grow exponentially in the input size of the problem, thus not allowing traditional methods to scale to large networks. Moreover, the design of good heuristics or approximation algorithms for network flow problems often requires significant specialized knowledge and trial-and-error.

In this paper, we argue that data-driven strategies have the potential to automate this challenging, tedious process, and learn efficient algorithms without compromising optimality. To do so, we

D. Gammelli et al., Graph Reinforcement Learning for Network Control via Bi-Level Optimization (Extended Abstract). Presented at the First Learning on Graphs Conference (LoG 2022), Virtual Event, December 9–12, 2022.

propose a graph network-based reinforcement learning framework that can handle a wide variety of network control problems. Specifically, we introduce a bi-level formulation that leads to dramatically improved scalability and performance by combining the strengths of mathematical optimization and learning-based approaches.

## 2    Problem Setting: Dynamic Network Control

To outline our problem formulation, we first define the linear problem, which is a classic convex problem formulation. We will then define a nonlinear, dynamic, non-convex problem setting that better corresponds to real-world instances. Much of the classical flow control literature and practice substitute the former linear problem for the latter nonlinear problem to yield tractable optimization problems [15–17]; we leverage the linear problem as an important algorithmic primitive. We consider the control of $N_c$ *commodities* on graphs, for example, vehicles in a transportation problem. A graph $\mathcal{G} = \{\mathcal{V}, \mathcal{E}\}$ is defined as a set $\mathcal{V}$ of $N_v$ nodes, and a set $\mathcal{E}$ of $N_e$ ordered pairs of nodes $(i, j)$ called edges, each described by a traversal time $t_{ij}$. We use $\mathcal{N}^+(i), \mathcal{N}^-(i) \subseteq \mathcal{V}$ for the set of nodes having edges pointing away from or toward node $i$, respectively. We use $s_i^t(k) \in \mathbb{R}$ to denote the quantity of commodity $k$ at node $i$ and time $t$[1].

**The Linear Network Control Problem.**    Within the linear model, our commodity quantities evolve in time as

$$s_i^{t+1} = s_i^t + f_i^t + e_i^t, \quad \forall i \in \mathcal{V} \tag{1}$$

where $f_i^t$ denotes the change due to flow of commodities along edges and $e_i^t$ denotes the change due to exchange between commodities at the same graph node. We refer to this expression as the *conservation of flow*. We also accrue money as

$$m^{t+1} = m^t + m_f^t + m_e^t, \tag{2}$$

where $m_f^t, m_e^t \in \mathbb{R}$ denote the money gained due to flows and exchanges respectively. Money can also be replaced with any other form of scalar reward, although it may be subject to e.g. non-negativity constraints and thus is different from the notion of reward in the RL problem. Our overall problem formulation will typically be to control **flows** and **exchanges** so as to maximize money over one or more steps subject to additional **constraints** such as, e.g., flow limitations through a particular edge.

**Flows.**    We will denote flows along edge $(i, j)$ with $f_{ij}^t(k)$. From these flows, we have

$$f_i^t = \sum_{j \in \mathcal{N}^-(i)} f_{ji}^t - \sum_{j \in \mathcal{N}^+(i)} f_{ij}^t, \quad \forall i \in \mathcal{V} \tag{3}$$

which is the net flow (inflows minus outflows). As discussed, associated with each flow is a cost $m_{ij}^t(k)$. Note that given this formulation, the total cost for all commodities can be written as $m_{ij}^t \cdot f_{ij}^t = (m_{ij}^t)^\top f_{ij}^t$. Thus, we can write the total flow cost at time $t$ as

$$m_f^t = \sum_{i \in \mathcal{V}} \left( \sum_{j \in \mathcal{N}^-(i)} m_{ji}^t \cdot f_{ji}^t - \sum_{j \in \mathcal{N}^+(i)} m_{ij}^t \cdot f_{ij}^t \right). \tag{4}$$

**Exchanges.**    To define our exchange relations and their effect on commodity quantities and costs, we will write the effect which exchanges have on money for each node; we write this as $m_i^t$. Thus, we have $m_e^t = \sum_{i \in \mathcal{V}} m_i^t$. The exchange relation takes the form

$$\begin{bmatrix} e_i^t \\ m_i^t \end{bmatrix} = E_i^t w_i^t \tag{5}$$

where $E_i^t \in \mathbb{R}^{N_c+1 \times N_e(i)}$ is an exchange matrix and $w \in \mathbb{R}^{N_e(i)}$ are the weights for each exchange. Each column in this exchange matrix denotes an (exogenous) exchange rate between commodities; for example, for $i$'th column $[-1, 1, 0.1]^\top$, one unit of commodity one is exchanged for one unit of commodity two plus $0.1$ units of money. Thus, choice of exchange weights $w_i^t$ uniquely determines exchanges $e_i^t$ and money change due to exchanges, $m_e^t$.

---

[1]We consider several reduced views over these quantities, and maintain several notational rules. We write $s_i^t \in \mathbb{R}^{N_c}$ to denote the vector of all commodities; we write $s^t(k) \in \mathbb{R}^{N_v}$ to denote the vector of commodity $k$ at all nodes; we write $s_i(k) \in \mathbb{R}^T$ to denote commodity $k$ at node $i$ for all times $t$. We can also apply any combination of these notation rules, yielding for example $s \in \mathbb{R}^{T \times N_c \times N_v}$.

**Linear Constraints.** We may impose additional (linear) constraints on the problem beyond the conservation of flow we have discussed so far. There are a few common examples that we may use in several applications. A common constraint is non-negativity of commodity values, which we may express as

$$s_i^t \geq 0, \quad \forall i, t. \tag{6}$$

Note that this inequality is defined element-wise. A similar constraint can be defined for money. We may also impose constraints on flows and exchanges; thus, we may for example limit the flow of all commodities through a particular edge via

$$\sum_{k=1}^{N_c} f_{ij}^t(k) \leq \overline{f}_{ij}^t \tag{7}$$

where this sum could also be weighted per-commodity. These linear constraints are only a limited selection of some common examples; the space of possible constraints is extremely general and the particular choice of constraints is problem-specific.

**The Nonlinear Dynamic Network Control Problem.** The previous subsection presented a linear problem formulation that yields a convex optimization problem for the decision variables—the chosen flow and exchange values. However, the formulation is limited by the assumption of linearity, thus lacking in the characterization of a number of elements typical of real-world systems (please refer to Appendix A for a more complete treatment). Crucially, these nonlinear, time-varying, stochastic, or unknown elements lead to severe difficulties in applying the convex formulation derived in the previous subsection. A common approach is to solve a linearized version of the nonlinear problem at each timestep, which is a form of model predictive control (MPC), although this essentially discards some elements of the problem to achieve computational tractability. In this paper, we focus on solving the nonlinear problem (reflecting real, highly general problem statements) via a bilevel optimization approach, wherein the linear problem (which has been shown to be extremely useful in practice) is used as an inner control primitive.

## 3 Methodology: The Bi-Level Formulation

In this section we describe the bi-level formulation that is the primary contribution of this paper. We further introduce a more formal Markov decision process (MDP) for our problem setting, together with a discussion on practical elements for real-world problem formulations in Appendix B.

**The Bi-Level Formulation.** We consider a discounted infinite-horizon MDP $\mathcal{M} = (\mathcal{S}, \mathcal{A}, P, R, \gamma)$. Here, $s^t \in \mathcal{S}$ is the state and $a^t \in \mathcal{A}$ is the action space, both at time $t$. The state in this setting is commodity values at nodes, as well as other available information; actions corresponds to aforementioned decision variables. The dynamics, $P : \mathcal{S} \times \mathcal{A} \times \mathcal{S} \to [0, 1]$ are probabilistic, with $P(s^{t+1} \mid s^t, a^t)$ denoting a conditional distribution over $s^{t+1}$. The reward function $R : \mathcal{S} \times \mathcal{A} \to \mathbb{R}$ is real-valued, and not limited to strictly positive or negative rewards. Finally, we write the discount factor as $\gamma$ as is typical in the infinite-horizon RL formulation, although it is straightforward to instead consider a finite-horizon setting. Please refer to Appendix B.1 for further treatment of the MDP.

The overall goal of the reinforcement learning problem setting is to find a policy $\tilde{\pi}^* \in \tilde{\Pi}$ (where $\tilde{\Pi}$ is the space of realizable Markovian policies) such that $\tilde{\pi}^* \in \arg\max_{\tilde{\pi} \in \tilde{\Pi}} \mathbb{E}_\tau \left[ \sum_{t=0}^\infty \gamma^t R(s^t, a^t) \right]$, where $\tau = (s^0, a^0, s^1, a^1, \ldots)$ denotes the trajectory of states and actions. This policy formulation requires specifying a distribution over all flow/exchange actions, which may be an extremely large space. We instead consider a bi-level formulation

$$\pi^* \in \arg\max_{\pi \in \Pi} \mathbb{E}_\tau \left[ \sum_{t=0}^\infty \gamma^t R(s^t, a^t) \right] \qquad \text{s.t. } a^t = \text{LCP}(\hat{s}^{t+1}, s^t) \tag{8}$$

where we consider a stochastic policy $\pi(\hat{s}^{t+1} \mid s^t)$, which maps from the current state to a *goal next state* (or subset of the state, such as commodity values only). This goal next state is used in the linear control problem (LCP$(\cdot, \cdot)$), which leverages a (slightly modified) one-step version of the linear problem formulation of Section 2 to map from desired next state to action. Thus, the resulting formulation is a bi-level optimization problem, whereby the policy $\tilde{\pi}$ is the composition of the policy $\pi(\hat{s}^{t+1} \mid s^t)$ and the solution to the linear control problem. Specifically, given a sample of $\hat{s}^{t+1}$ from

**Table 1:** Average performance across multiple environments over 100 test episodes

|  |  | Random | MLP-RL | GCN-RL | GAT-RL | MPNN-RL (ours) | Oracle |
|---|---|---|---|---|---|---|---|
| 2-hops | Avg. Reward | 63 | 387 | 201 | 146 | **576** | 642 |
|  | % Oracle | 9.9% | 60.2% | 31.3% | 22.9% | **89.7%** | - |
| 3-hops | Avg. Reward | 1013 | 1084 | 1385 | 1257 | **1803** | 2014 |
|  | % Oracle | 50.3% | 53.8% | 68.7% | 62.4% | **89.5%** | - |
| 4-hops | Avg. Reward | 2033 | 2185 | 2303 | 2198 | **2807** | 3223 |
|  | % Oracle | 63.1% | 67.8% | 71.4% | 68.2% | **87.1%** | - |
| Dyn tt | Avg. Reward | -546 | -18 | 437 | 400 | **2306** | 2327 |
|  | % Oracle | -23.4% | -0.7% | 18.7% | 17.1% | **99.1%** | - |
| Dyn top | Avg. Reward | 810 | N/A | 1016 | 827 | **1599** | 1904 |
|  | % Oracle | 42.5% | N/A | 53.4% | 43.4% | **83.9%** | - |
| Capacity | Avg. Reward | 1495 | 1498 | 1557 | 1503 | **2145** | 2389 |
|  | % Oracle | 62.6% | 62.7% | 65.2% | 62.9% | **89.8%** | - |
|  | Success Rate | 82% | 82% | 87% | 80% | **87%** | 88% |
| Multi-com | Avg. Reward | 2191 | 4045 | 3278 | 3206 | **6986** | 9701 |
|  | % Oracle | 22.5% | 41.7% | 33.8% | 33.0% | **72.0%** | - |

the stochastic policy, we select concrete flow and exchange actions by solving the linear control problem, defined as

$$\underset{a^t}{\arg\min} \quad d(\hat{s}^{t+1}, s^{t+1}) - R(s^t, a^t) \tag{9a}$$

$$\text{s.t.} \quad \text{Conservation of flow (1); Net flow (3); Flow reward (4);} \tag{9b}$$

$$\text{Exchange conditions (5); Other constraints, e.g. (6) or (7)} \tag{9c}$$

where $d(\cdot, \cdot)$ is a chosen convex metric which penalizes deviation from the desired next state. The resultant problem—consisting of a convex objective subject to linear constraints—is convex and thus may be easily and inexpensively solved to choose actions $a^t$, even for very large problems.

As is standard in reinforcement learning, we will aim to solve this problem via learning the policy from data. This may be in the form of online learning [18] or via learning from offline data [19]. There are large bodies of work on both problems, and our presentation will generally aim to be as-agnostic-as-possible to the underlying reinforcement learning algorithm used. Of critical importance is the fact that the majority of reinforcement learning algorithms use likelihood ratio gradient estimation (typically referred to as the REINFORCE gradient estimator in RL [20]), which does not require path-wise backpropagation through the inner problem.

We also note that our formulation assumes access to a model (the linear problem) that is a reasonable approximation of the true dynamics over short horizons. This short-term correspondence is central to our formulation: we exploit exact optimization when it is useful, and otherwise push the impacts of the nonlinearity over time in the learned policy. We assume this model is known in our experiments, but it could be identified independently. Please see Appendix C.1, C.2, and C.4 for a broader discussion.

**Network Architecture.** To exploit the network structure of the problem we introduce a policy graph neural network architecture based on message passing neural networks [21] (Appendix B.2). As introduced in this section, the goal of RL is to learn a stochastic policy $\pi(\hat{s}^{t+1} \mid s^t)$ mapping to goal next states. Concretely, to obtain a valid probability density over next states, we define the output of our policy network to represent the concentration parameters $\alpha \in \mathbb{R}_+^{N_v}$ of a Dirichlet distribution, such that $\hat{s}^{t+1} \sim \text{Dir}(\hat{s}^{t+1}|\alpha)$, although alternate output formulations are possible.

## 4 Experiments

In this section, we compare against a number of benchmarks on an instance of network control with great real-world impact: the *minimum cost flow problem*. Within this context, the goal is to control commodities so to move them from one or more source nodes to one or more sink nodes, in the minimum time possible. Appendix E provides further details on both benchmarks and environments.

**Minimum cost flow through message passing.** In this first experiment, we consider 3 different environments (Fig. 1), such that different topologies enforce a different number of *required hops of message passing* between source and sink nodes to select the best path. Results in Table 1 (*2-hop*, *3-hop*, *4-hop*) show how MPNN-RL is able to achieve at least $87\%$ of oracle performance. Table 1 further shows how agents based on graph convolutions (i.e., GCN [22], GAT [23]) fail to learn an effective flow optimization strategy. As in Xu et al. [24], we argue in favor of the *algorithmic alignment* between the computational structure of MPNNs and the kind of computations needed to solve traditional network optimization problems (see Appendix C.3 for further discussion).

**Dynamic traversal times.** In this experiment, we define time-dependent traversal times. In Fig. 2 and Table 1 (*Dyn tt*) we measure results on a dynamic network characterized by two *change-points*, i.e., time steps where the optimal path changes because of a change in traversal times. Results show how the proposed MPNN-RL is able to achieve above $99\%$ of oracle performance.

**Dynamic topology.** In this experiment we assume a time-dependent topology, i.e., nodes and edges can be dropped or added during an episode. This case is substantially different from what most traditional approaches are able to handle: the locality of MPNN agents together with the one-step implicit planning of RL, enable our framework to deal with multiple graph configurations during the same episode. Fig. 3 and Table 1 (*Dyn top*) show how MPNN-RL achieves 83.9% of oracle performance clearly outperforming the other benchmarks. Crucially, these results highlight how agents based on MLPs result in highly inflexible network controllers, thus limited to a fixed topology.

**Capacity constraints.** In this experiment, we relax the assumption that capacities $\bar{f}_{ij}$ are always able to accommodate any flow on the graph. Compared to previous sections, the lower capacities introduce the possibility of infeasible states. To measure this, the *Success Rate* computes the percentage of episodes which have been terminated successfully. Results in Table 1 (*Capacity*) highlight how MPNN-RL is able to achieve 89.8% of oracle performance while being able to successfully terminate 87% of episodes. Qualitatively, Fig. 4 shows a visualization of the policy for a specific test episode. The plots show how the MPNN-RL is able to learn the effects of capacity on the optimal strategy by allocating flow to a different node when the corresponding edge is approaching its capacity limit.

**Multi-commodity.** In this scenario, we extend the current architecture to deal with multiple commodities and source-sink combinations. Results in Table 1 (*Multi-com*) and Fig. 5 show how MPNN-RL is able to effectively recover distinct policies for each policy head, thus being able to operate successfully multi-commodity flows within the same network.

**Computational analysis.** We study the computational cost of MPNN-RL compared to MPC-based solutions. As shown in Fig. 6, we compare the time necessary to compute a single network flow decision. We do so across varying dimensions of the underlying graph, ranging from 10 up to 400 nodes. As verified by this experiment, learning-based approaches exhibit computational complexity linear in the number of nodes and graph connectivity, without significant decay in performance.

## 5 Outlook and Limitations

Research in network flow models, in both theory and practice, is largely scattered across the control, management science, and optimization literature, potentially hindering scientific progress. In this work, we propose a general framework that could enable learning-based approaches to help address the open challenges in this space: handling nonlinear dynamics and scalability, among others. In the hope of fostering a unification of tools among the reinforcement learning and network control communities, we aimed to (i) maintain the narration as-agnostic-as-possible, and (ii) showcase the extreme versatility of our framework through numerous controlled experiments. However, what we present here should be considered as, in our opinion, exciting preliminary results aiming to gather more traction among the ML community towards the solution of hugely impactful real-world problems in the field of network control. Crucially, before being able to consider learning-based frameworks as a concrete alternative to current standards, we believe this research opens several promising future directions for the extension of these concepts to large-scale applications.

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

## A   Dynamic Network Control

In this section we make concrete our discussion on nonlinear problem formulations for the dynamic network flow problem.

**Elements breaking the linearity assumptions.**   Real-world systems are characterized by many factors that cannot be reliably modelled through the linear problem described in Section 2. In what follows, we discuss a (non-exhaustive) list of factors potentially breaking such linearity assumptions:

- **Stochasticity.** Various stochastic elements can impact the problem. Commodity transitions in the previous section were defined as being deterministic; in practice in many problems, there are elements of stochasticity to these transitions. For example, random demand may reduce supply by an unpredictable amount; vehicles may be randomly added in a transportation problem; and packages may be lost in a supply chain setting. In addition to these state transitions, constraints may be stochastic as well: flow times or edge capacities may be stochastic, as when a road is shared with other users, or costs for flows and exchange may be stochastic.

- **Nonlinearity.** Various elements of the state evolution, constraints, or cost function may be nonlinear. The objective may be chosen to be a risk-sensitive or robust metric applied to the distribution of outcomes, as is common in financial problems. The state evolution may have natural saturating behavior (e.g. automatic load shedding). Indeed, many real constraints will have natural nonlinear behavior.

- **Time-varying costs and constraints.** Similar to the stochastic case, various quantities may be time-varying. However, it is possible that they are time-varying in a structured way, as opposed to randomly. For example, demand for transportation may vary over the time of day, or purchasing costs may vary over the year.

- **Unknown dynamics elements.** While not a major focus of discussion in the paper up to this point, elements of the underlying dynamics may be partially or wholly unknown. Our reinforcement learning formulation is capapble of addressing this by learning policies directly from data, in contrast to standard control techniques.

## B   Methodology

In this section we discuss the full MDP formulation (including defining state and action spaces) and discuss algorithmic details.

### B.1   The Dynamic Network MDP

The problem setting for the full, dynamic network problem is best formulated, in the general case, as a partially-observed MDP. We will present it as a Markovian decision process (fully-observed), where the choice of input features beyond commodity values are chosen by the user; discussion on strategies for better handling partial-observability are presented later in this section.

We consider a discounted infinite-horizon MDP $\mathcal{M} = (\mathcal{S}, \mathcal{A}, P, R, \gamma)$. Here, $s^t \in \mathcal{S}$ is the state and $a^t \in \mathcal{A}$ is the action space, both at time $t$. The dynamics, $P : \mathcal{S} \times \mathcal{S} \times \mathcal{A} \rightarrow [0, 1]$ are probabilistic, with $P(s^{t+1} \mid s^t, a^t)$ denoting a conditional distribution over $s^{t+1}$. The reward function $R : \mathcal{S} \times \mathcal{A} \rightarrow \mathbb{R}$ is real-valued, and not limited to strictly positive or negative rewards. Finally, we write the discount factor as $\gamma$ as is typical in the infinite-horizon RL formulation, although it is straightforward to instead consider a finite-horizon setting.

**State and state space.**   We will, generally, define the state to contain enough information to yield "good" Markovian policies. More formally, real-world network control problems are typically highly partially-observed; many features of the world impact the state evolution. However, a small number of features are typically of primary importance, and the impact of the other partially-observed elements can be modeled as stochastic disturbances.

For our bi-level formulation, there are some state elements that are required. Our formulation requires, at each timestep, the commodity values $s^t$. Furthermore, the constraint values are required, such as costs, exchange rates, flow capacities, etc. If the graph topology is time-varying, the connectivity at time $t$ is also critical. The state values to fully define the one-step linear control problem are the only state elements which are required. We refer to these constraint values as edge state elements. More precisely, the state elements we have discussed so far are either properties of the graph nodes

(commodity values) or of the edges (such as flow constraints). This difference is of critical importance in our graph neural network architecture.

In addition to these state elements, additional information may be incorporated. Generally, the choice of state elements will depend on the information available to a system designer (what can be measured) and will depend on the particular problem setting. Possible examples of further state elements include forecasts of prices, exchange rates, or flow constraints at future times; exchanges rates, for example, include notions of demand or supply. We note that such forecasts are almost always available, as they are necessary for solving the multi-step planning problem.

**Action and action space.** As discussed in Section 2, the action is defined as all flows and exchange weights at all nodes/edges, $a^t = (f^t, w^t)$. In our bi-level formulation, we compute $a^t$ by replacing a single policy mapping from states to actions with two nested policies, whereby the goal next state $s^{t+1}$ acts as an intermediate variable, thus avoiding the parametrization of an extremely large action space, e.g., flows over edges in a graph. Specifically, the goal next state is fed into a linear control problem (LCP) to obtain the edge-wise flows that would best match the desired next state. Thus, the policy for the original problem is defined as a composition of policies.

**Dynamics.** The dynamics of the MDP, $P$, describe the evolution of state elements. We split our discussion in to two parts: the dynamics associated with the commodity time evolution and the dynamics of the non-commodity elements.

The commodity dynamics are assumed to be reasonably well-modeled by the conservation of flow, (1), subject to the constraints; this forms the basis of the bi-level approach we describe in the next subsection. The primary element not included in the conservation of flow expression is possible stochasticity. For example, in transportation problems, vehicles may randomly drop out of service.

The non-commodity dynamics are assumed to be substantially more complex. For example, prices to buy or sell (reflected in exchange rates) may have a complex dependency on past sales, current demand, and current supply (commodity values), as well as random exogenous factors. Thus, we place few assumptions on the evolution of non-commodity dynamics, and assume that current values are measurable.

**Reward.** Throughout the paper, we will assume our full reward is the total discounted money earned over the (infinite) problem duration. This results in a stage-wise reward function that corresponds simply to the money earned in that time period, or

$$R(s^t, a^t) = m_e^t + m_f^t. \tag{10}$$

Note that the sum of rewards to time $t$ is exactly $m^t - m^0$, which corresponds to the money earned.

It is typical in economics and finance to consider concave utility functions or risk metrics as opposed to the exact return [25, 26]. However, this reward structure does not result in a simple stage-wise reward decomposition as in the linear case. Thus, while addressing this concavity is important, we do not address it in this work.

### B.2 Network Architecture and RL Details

In this section we introduce the basic building blocks of our graph neural network architecture. Let us define with $\mathbf{x}_i \in \mathbb{R}^{D_\mathbf{x}}$ and $\mathbf{e}_{ji} \in \mathbb{R}^{D_\mathbf{e}}$ the $D_\mathbf{x}$-dimensional vector of node features of node $i$ and the $D_\mathbf{e}$-dimensional vector of edge features from node $j$ to node $i$, respectively.

We define the update function of node features through the following message passing neural network (MPNN):

$$\mathbf{x}_i^{(k)} = \max_{j \in \mathcal{N}^-(i)} f_\theta \left( \mathbf{x}_i^{(k-1)}, \mathbf{x}_j^{(k-1)}, \mathbf{e}_{ji} \right), \tag{11}$$

where $k$ indicates the $k$-th layer of message passing in the GNN with $k = 0$ indicating raw environment features, i.e., $\mathbf{x}_i^{(0)} = \mathbf{x}_i$, and where we use the element-wise $\max$ operator as aggregation function in our proposed graph-network.

We note that this network architecture can be used to define both policy and value function estimator, depending on the reinforcement learning algorithm of interest (e.g., actor-critic [27], value-based [28], etc.). As an example, in our implementation, we define two separate decoder architectures for the actor and critic networks of an Advantage Actor Critic (A2C) [29] algorithm. For the actor, we define the output of our policy network to represent the concentration parameters $\alpha \in \mathbb{R}_+^{N_v}$ of a

Dirichlet distribution, such that $\mathbf{a}_t \sim \text{Dir}(\mathbf{a}_t | \alpha)$, and where the positivity of $\alpha$ is ensured by a Softplus nonlinearity. On the other hand, the critic is characterized by a *global* sum-pooling performed after $K$ layers of MPNN. In this way, the critic computes a single value function estimate for the entire network by aggregating information across all nodes in the graph.

**Handling dynamic topologies.** A defining property of our framework is its ability to deal with time-dependent graph connectivity (e.g., edges or nodes are added/dropped during the course of an episode). Specifically, our framework achieves this by (i) considering the problem as a one-step decision-making problem, i.e., avoiding the dependency on potentially unknown future topologies, and (ii) exploiting the capacity of GNNs to handle diverse graph topologies. Crucially, no matter the current state of the graph, GNN-based agents are capable of computing a goal next state for the network, which will then be converted into actionable flow decisions by the LCP.

**Exploration.** In practice, we choose large penalty terms $d(\cdot, \cdot)$ to minimize greediness. However early in training, randomly initialized penalty terms can harm exploration. In practice, we found it was sufficient to down-weight the penalty term early in training. As such, the inner action selection is biased toward short-term rewards, resulting in greedy action selection. However, there are many further possibilities for exploiting random penalty functions to induce exploration, which we discuss in the next section.

**Integer-valued flows.** For several problem settings, it is desirable that the chosen flows be *integer-valued*. For example, in a transportation problem, we may wish to allocate some number of vehicles, which can not be infinitely sub-divided [5, 30]. There are several ways to introduce integer-valued constraints to our framework. First, we note that because the RL agent is trained through policy gradient—and thus we do not require a differentiable inner problem—we can simply introduce integer constraints into the lower-level problem[2]. However, solving integer-constrained problems is typically expensive in practice. An alternate solution is to simply use a heuristic rounding operation on the output of the inner problem. Again, because of the choice of gradient estimator, this does not need to be differentiable. Moreover, the RL policy learns to adapt to this heuristic clipping. Thus, we in general recommend this strategy as opposed to directly imposing constraints in the inner problem.

## C Discussion and Algorithmic Components

In this section we discuss various elements of the proposed framework, highlight correspondences and design decisions, and discuss component-level extensions.

### C.1 Distance metric as value function

The role of the distance metric (and the generated goal next state) is to capture the value of future reward in the greedy one-step inner optimization problem. This is closely related to the value function in dynamic programming and reinforcement learning, which in expectation captures the sum of future rewards for a particular policy. Indeed, under moderate technical assumptions, our linear problem formulation with stochasticity yields convex expected cost-to-go (the negative of the value) [32, 33].

There are several critical differences between our penalty term and a learned value function. First, a value function in a Markovian setting for a given policy is a function solely of state. For example, in the LCP, a value function would depend only on $s^{t+1}$. In contrast, our value function depends on $\hat{s}^{t+1}$, which is the output of a policy which takes $s^t$ as an input. Thus, the penalty term is a function of both the current and predicted next state. Given this, the penalty term is better understood as a local approximation of the value function, for which convex optimization is tractable, or as a form of state-action value function with a reduced action space (also referred to as a Q function).

The second major distinction between the penalty term and a value function is particular to reinforcement learning. Value functions in modern RL are typically learned via minimizing the Bellman residual [18], although there is disagreement on whether this is a desirable objective [34]. In contrast, our policy is trained directly via gradient descent on the total reward (potentially incorporating value function control variates). Thus, the objective for this penalty method is better aligned with maximizing total reward.

---

[2]Note that several problems exhibit a *total unimodularity* property [31], for which the relaxed integer-valued problem is tight.

## C.2 Beyond a single-step inner problem

Our formulation so far has considered a bi-level formulation in which the RL policy outputs a desired state at the *next* timestep, $\hat{s}^{t+1}$; this is then used in the lower-level problem to select actions. There are two relaxations to this procedure that can be incorporated here.

First, the RL policy can output any future state, and direct optimization can happen for any horizon. We may parameterize the RL policy to return $\hat{s}^{t+k}$ for $k \geq 1$. Given this, a multi-step optimization problem may be solved using the linear model. The potential risk to this approach is the linear (in horizon) growth in variables for the inner problem, and poor agreement between the linear model and the nonlinear model. This presents a strict generalization of our proposed method. The primary reason we have not considered the multi-step formulation as the primary algorithm of this paper is that it requires modeling the dynamics of the non-commodity state variables. For example, this model requires forecasting all constraint values, whereas our one-step formulation requires only knowledge at the current timestep. Forecasting of constraint values is closely linked to questions of (persistent) feasibility, which we do not consider in detail in this paper.

Second, stochasticity may be directly integrated into the lower-level problem. The standard formulation for stochastic model predictive control (or stochastic multi-stage optimization) is the scenario formulation [35], in which a tree of outcomes is constructed via sampling noise realizations[3]. Within the one-step bi-level formulation, sampling $N_n$ noise realizations results in $N_n$ values of the next state, $s_i^{t+1}$, $i = 1, \ldots, N_n$ within the inner problem. The empirical mean loss

$$\mathbb{E}_{s^{t+1}}[d(\hat{s}^{t+1}, s^{t+1})] - R(s^t, a^t) \approx \frac{1}{N_n} \sum_{i=1}^{N_n} d(\hat{s}^{t+1}, s_i^{t+1}) - R(s^t, a^t) \tag{12}$$

can then be minimized. We emphasize that the actions are the same for each noise realization—this is the so-called *non-anticipativity* constraint. This formulation, for one step, does not meaningfully increase the number of decision variables, although will result in increased computational complexity. More importantly, multi-step optimization within the scenario tree approach yields exponential growth in the number of decision variables, which will rapid result in intractability. We refer the reader to [35] for more details on scenario-based stochastic optimization.

## C.3 Graph neural networks for network optimization

In this work, we argue that GNNs represent a natural choice for graph optimization problems because of three main properties:

First, permutation invariance, or, more specifically, GNNs represent a class of functions for which the output is independent of the node ordering. Crucially, non-permutation invariant computations would consider each ordering as fundamentally different and thus require an exponential number of input/output training examples before being able to generalize.
Second, locality of the operator. GNNs typically express a local parametric filter (e.g., convolution operator) which enables the same neural network to be applied to graphs of varying size and connectivity: a property of fundamental importance for network flow optimization problems specifically, and real-world problems of economic importance more generally, which will be dynamic or frequently re-configured, and it is desirable to be able to use the same policy without re-training.
Lastly, alignment with the computations used for network optimization problems. The concept of algorithmic alignment refers to the fact that despite many neural network architectures have the *capacity* to represent a wide range of algorithms, not all networks are able to actually *learn* these algorithms. Intuitively, a network may learn and generalize better if it is able to represent a function (algorithm) "more easily." A notable example of this in the context of supervised learning is the relation between MLPs and CNNs in computer vision—where MLPs are theoretically universal approximators yet struggle to achieve satisfying performance on most vision tasks. The difference in results of MLP-RL in Table 1 (*2-hops*) compared to Table 1 (*3-hops*, *4-hops*) further confirms these concepts, whereby the smaller dimensionality of the 2-hops environment leads to a smaller solution space for the MLPs, which are able to converge to relatively good policies. On the other hand, the 3-hops and 4-hops environments are characterized by a significant increase in the number of edges and nodes, leading to a more challenging search for solutions in policy-space.

---

[3]We note that non-sampling strategies such as moment-matching formulations are also possible, although we will not discuss these methods herein.

## C.4 Computational efficiency

Consider solving the full nonlinear control problem via direct optimization over a finite horizon ($T$ timesteps), which corresponds to a model predictive control [36] formulation. How many total actions must be selected? The number of possible flows for a fully dense graph (worst case) is $N_v(N_v - 1)$. In addition to this, there are $\sum_{i \in \mathcal{V}} N_e(i)$ possible exchange actions; if we assume $N_e$ is the same for all nodes, this yields $N_v N_e$ possible actions. Finally, we have $N_c$ commodities. Thus, the worst-case number of actions to select is $TN_cN_v(N_v + N_e - 1)$; it is evident that for even moderate choices of each variable, the complexity of action selection in our problem formulation quickly grows beyond tractability.

While moderately-sized problems may be tractable within the direct optimization setting, we aim to incorporate the impacts of stochasticity, nonlinearity, and uncertainty, which typically results in non-convexity. The reinforcement learning approach, in addition to being able to improve directly from data, reduces the number of actions required to those for a single step. If we were to directly parameterize the naive policy that outputs flows and exchanges, this would correspond to $N_cN_v(N_v + N_e - 1)$ actions. For even moderate values of $N_c, N_v, N_e$, this can result in millions of actions. It is well-known that reinforcement learning algorithms struggle with high dimensional action spaces [37], and thus this approach is unlikely to be successful. In contrast, our bi-level formulation requires only $N_c$ actions for the learned policy, while additionally leveraging the beneficial inductive biases over short time horizons.

# D   Related Work

Bi-level optimization—in which one optimization problem depends on the solution to another optimization problem, and are thus nested—has recently become an important topic in machine learning, reinforcement learning, and control [38–44]. Of particular relevance to our framework are methods that combine principled control strategies with learned components in a hierarchical way. Examples include using LQR control in the inner problem with learnable cost and dynamics [41, 45, 46]; learning sampling distributions in planning and control [47–49]; or learning optimization strategies or goals for optimization-based control [50, 51].

Numerous strategies for learning control with bi-level formulations have been proposed. A simple approach is to insert intermediate goals to train lower-level components, such as imitation [47]. This approach is inherently limited by the choice of the intermediate objective; if this objective does not strongly correlate with the downstream task, learning could emphasize unnecessary elements or miss critical ones. An alternate strategy, which we take in this work, is directly optimizing through an inner controller. A large body of work has focused on exploiting exact solutions to the gradient of (convex) optimization problems at fixed points [41, 46, 52]. This allows direct backpropatation through optimization problems, allowing them to be used as a generic component in a differentiable computation graph (or neural network). Our approach leverages likelihood ratio gradients (equivalently, policy gradient), an alternate zeroth-order gradient estimator [53]. This enables easy differentiation through lower-lever optimization problems without the technical details necessitated with fixed-point differentiation.

# E   Experiments

## E.1   Benchmarks

All RL modules were implemented using PyTorch [54] and the IBM CPLEX solver [55] for the optimization problem. In our experiments, we compare the proposed framework with the following methods:

**Heuristics.**   In this class of methods, we focus on measuring performance of simple, domain-knowledge-driven rebalancing heuristics.

1. *Random policy*: at each timestep, we sample the desired distribution from a Dirichlet prior with concentration parameter $\alpha = [1, 1, \ldots, 1]$. This benchmark provides a lower bound of performance by choosing desired goal states randomly.

**Learning-based.**   Within this class of methods, we focus on measuring how different architectures affect the quality of the solutions for the dynamic network control problem. For all methods, the A2C algorithm is kept fixed, thus the difference solely lies in the neural network architecture.

3. *MLP-RL*: both policy and value function estimator are parametrized by feed-forward neural networks. In all our experiments, we use two layers of 32 hidden unites and an output layer mapping to the output's support (e.g., a scalar value for the critic network). Through this comparison, we highlight the performance and flexibility of graph representations for network-structured data.

4. *GCN-RL*: In all our experiments, we use $K$ layers of graph convolution with 32 hidden units, with $K$ equal to the number of sink-to-source hops in the graph, and a linear output layer mapping to the output's support. See below for a broader discussion of graph convolution operators.

5. *GAT-RL*: In all our experiments, we use $K$ layers of graph attention with 32 hidden units, with $K$ equal to the number of sink-to-source hops in the graph, and single attention head. The output is further computed by a linear output layer mapping to the output's support. Together with GCN-RL, this model represents an approach based on graph convolutions rather than explicit message passing along the edges (as in MPNNs). Through this comparison, we argue in favor of explicit, pair-wise messages along the edges, opposed to sole aggregation of node features among a neighborhood. Specifically, we argue in favor of the alignment between MPNN and the kind of computations required to solve flow optimization tasks, e.g., propagation of travel times and selection of best path among a set of candidates (max aggregation).

6. *MPNN-RL*: ours. We use $K$ layers of MPNN of 32 hidden units as defined in Section B.2, with $K$ equal to the number of sink-to-source hops in the graph, and a linear output layer mapping to the output's support.

**MPC-based.** Within this class of methods, we focus on measuring performance of MPC approaches that serve as state-of-art benchmarks for the dynamic network flow problem.

5. *MPC-Oracle*: we directly optimize the flow using a standard formulation of MPC [56]. Notice that although the embedded optimization is a linear programming model, it may not meet the computation requirement of real-time applications (e.g., obtaining a solution within several seconds) for large scale networks. In this work, MPC is assumed to have access to future state elements (e.g., future traversal times, connectivity, etc.). Crucially, assuming knowledge of future state elements is equivalent to assuming oracle knowledge of the realization of all stochastic elements in the system. In other words, there is no uncertainty for the MPC (this is in contrast with RL-based benchmarks, that assume access only to *current* state elements). In our experiments, the benchmark with the "Oracle" MPC enables us to quantify the optimal solution for all environments, thus giving a sense of the optimality gap between the ground truth optimum and the solution achieved via RL.

### E.2 Environments

In what follows, we describe the properties of the environments used to train and evaluate our framework in order to make them reproducible and understandable. We select environment variables in a way to cover a wide enough range of possible scenarios, e.g., different traversal times and thus, different optimal actions.

- **Generalities**. As discussed in Section 4, the environments describe a dynamic minimum cost flow problem, whereby the goal is to let commodities flow from source to sink nodes in the minimum time possible (i.e., cost is equal to time). Formally, given a graph $\mathcal{G} = \{\mathcal{V}, \mathcal{E}\}$, the reward function across all environments is defined as:

$$R(s^t, a^t) = -\sum_{ij \in \mathcal{E}} f_{ij}^t t_{ij} + \lambda f_{\text{sink}}^t,$$

where $f_{ij}^t$ and $t_{ij}$ represent flow and traversal time along edge $(i, j)$ at time $t$, respectively, $f_{\text{sink}}^t$ is the flow arriving at all sink nodes at time $t$, and $\lambda$ is a weighting factor between the two reward terms. In our experiments, the resulting policy proved to be broadly insensitive to values of $\lambda$, with $\lambda \in [15, 30]$ typically being an effective range.

- **Minimum cost flow through message passing**. Given a single-source, single-sink network, we assume travel times to be constant over the episode and requirements (i.e., demand) to be sampled at each time step as $\rho = 10 + \psi_i$, $\psi_i \sim \text{Uniform}[-2, 2]$. Capacities $u_{ij}$ are fixed to a very high positive number, thus not representing a constraint in practice. Cost $m_{ij}$ is considered equal to the traversal time $t_{ij}$. An episode is assumed to have a duration of 30 time steps

and terminates when there is no more flow traversing the network. To present a variety of scenarios to the agent at training time, we sample random travel times for each new episode as $t_{ij} \sim \text{Uniform}[0, 10]$ and use the topologies shown in Fig. 1. In our experiments, we apply as many layers of message passing as hops from source to sink node in the graph, e.g., $K = 2$ and $K = 3$ in the 2-hops and 3-hops environment, respectively.

- **Dynamic traversal times**. To train our MPNN-RL, we select the 3-hops environment and generate travel times as follows for every episode: (i) sample random traversal times as $t_{ij} \sim \text{Uniform}[0, 10]$, (ii) for every time step, gradually change the traversal time as $t_{ij} = t_{ij} + \psi, \psi \sim \text{Uniform}[-1, 1]$.

- **Capacity constraints**. In this experiment, we focus on the 3-hops environment and assume a constant value $\bar{f}_{ij} = 20, \forall i, j \in \mathcal{V} : j \neq 7$ while we keep a high value for all the edges going into node 7 (i.e., the sink node) which would more easily generate infeasible scenarios. From an RL perspective, we add the following edge-level features:
    - Edge-capacity $\{\bar{f}_{ij}^t\}_{i,j \in \mathcal{V}}$ at the current time step $t$.
    - Accumulated flow $\{f_{ij}^t\}_{i,j \in \mathcal{V}}$ on edge $ij$

- **Multi-commodity**. Let $N_c$ define the number of commodities to consider, indexed by $k$. From an RL perspective, we extend the our proposed policy graph neural network to represent a $N_c$-dimensionsional Dirichlet distribution. Concretely, we define the output of the policy network to represent the $N_c \times N_v$ concentration parameters $\alpha \in \mathbb{R}_+^{N_c \times N_v}$ of a Dirichlet distribution over nodes for each commodity, such that $\mathbf{a}_t \sim \text{Dir}\{\mathbf{a}_t | \alpha\}$. In other words, to extend our approach to the multi-commodity setting, we define a multi-head policy network characterized by one head per commodity. In our experiments, we train our multi-head agent on the topology shown in Fig. 5 whereby we assume two parallel commodities: commodity A going from node 0 to node 10, and commodity B going from node 0 to node 11. We choose this topology so that the only way to solve the scenario is to discover distinct behaviours between the two network heads (i.e., the policy head controlling flow for commodity A needs to go up or it won't get any reward, and vice-versa for commodity B).

- **Computational analysis**. In this experiment, we generate different versions of the 3-hops environment, whereby different environments are characterized by intermediate layers with increasing number of nodes and edges. The results are computed by applying the pre-trained MPNN-RL agent on the original 3-hops environment (i.e., characterized by 8 nodes in the graph). In light of this, Figure 6 showcases a promising degree of transfer and generalization among graphs of different dimensions.

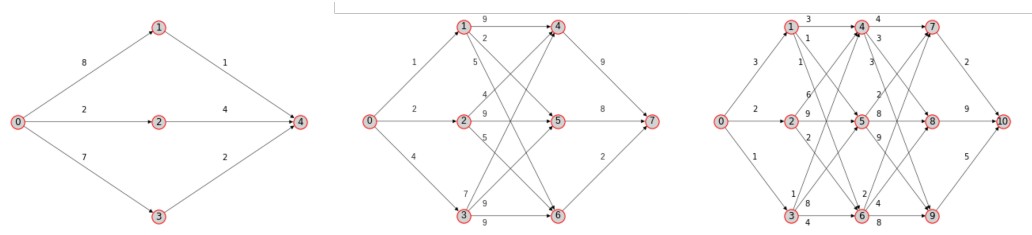

**Figure 1:** Graph topologies used for the message passing experiments: 2-hops (left), 3-hops (center), 4-hops (right). The source and sink nodes are represented by the left-most and right-most nodes, respectively. Values in proximity of the edges represent traversal times.

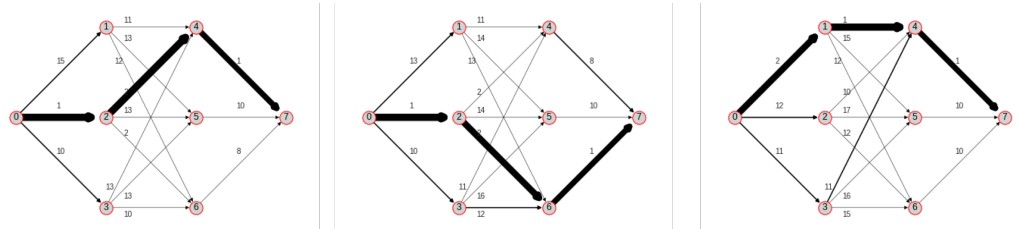

**Figure 2:** Visualization of a trained instance of MPNN-RL on an environment with dynamic traversal times. We simulate a scenario where the optimal path changes three times (left, middle, and right) over the course of an episode. Shaded edges represent actions induced by the MPNN-RL.

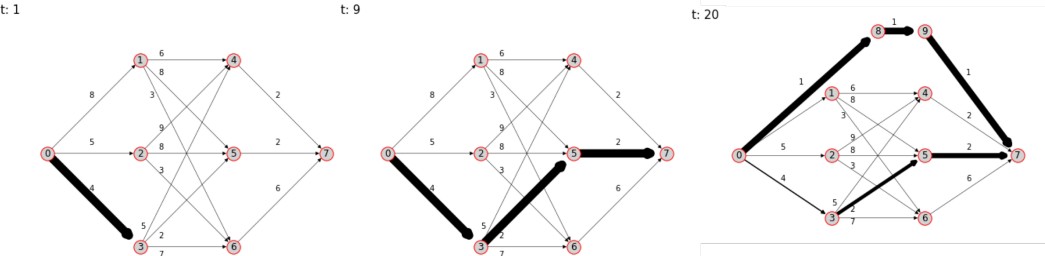

**Figure 3:** Visualization of a trained instance of MPNN-RL on an environment with dynamic topology. We simulate a scenario where the optimal path changes over the course of an episode because of the addition of a new path. Shaded edges represent actions induced by the MPNN-RL.

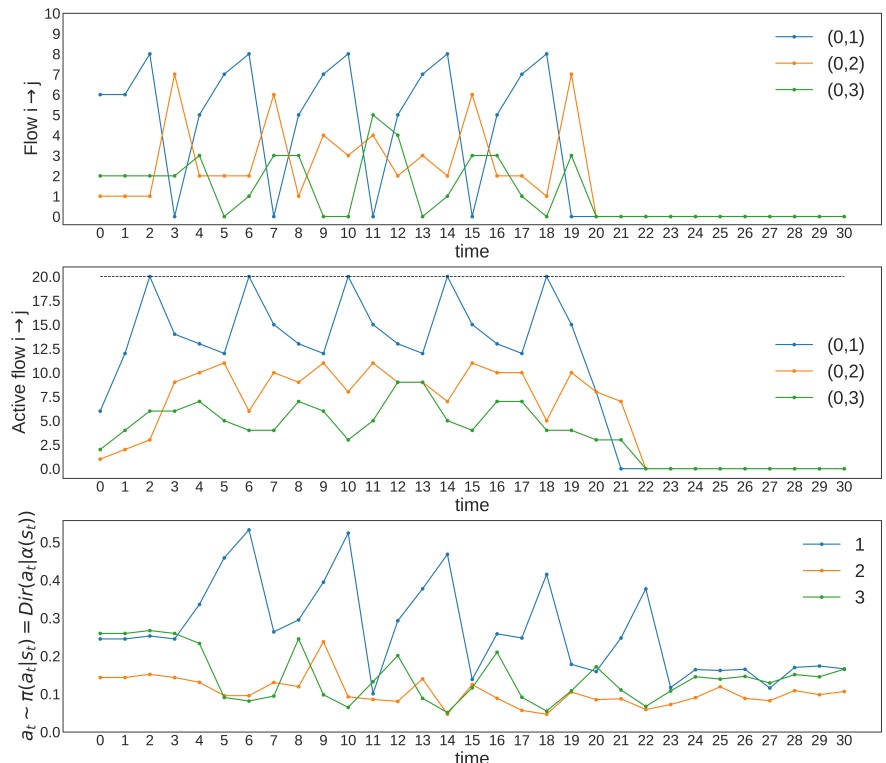

**Figure 4:** Visualization of the MPNN-RL policy on the capacity constrained environment. (Top) The resulting flow $f_{ij}$ on the edges $0 \to 1, 0 \to 2, 0 \to 3$. (Center) The accumulated flow on the same edges compared to the fixed capacity $\bar{f}_{ij} = 20$, represented as a dashed horizontal line. (Bottom) The desired distribution described by the MPNN-RL policy.

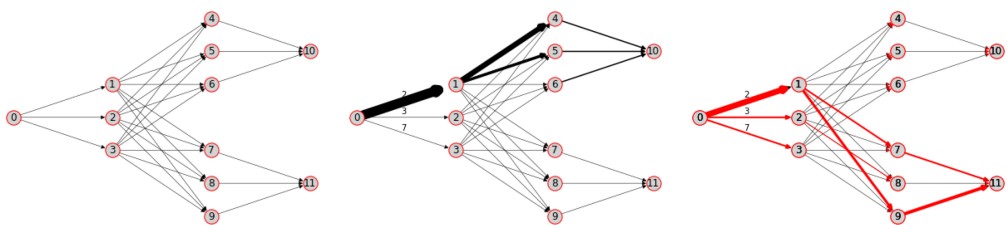

**Figure 5:** Visualization of the multi-commodity environment. (Left) The topology considered during our experiments. (Center) A visualization of the policy for the first commodity A. (Right) A visualization of the policy for the second commodity B.

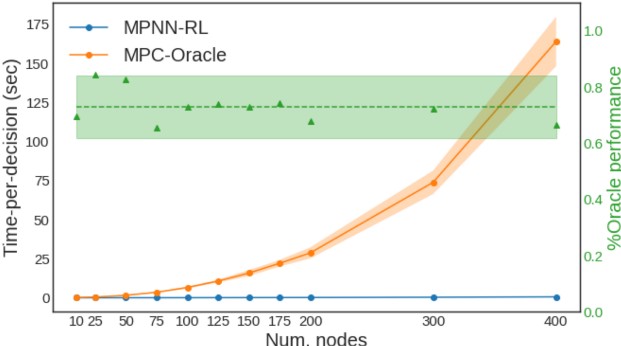

**Figure 6:** Comparison of computation times between learning-based (blue) and control-based (orange) approaches. Green triangles represent the percentage performance of our RL framework compared to the oracle model.

