# OpenReview forum: "Graph Reinforcement Learning for Network Control via Bi-Level Optimization"
_logconference.io/LOG/2022/Conference — LoG 2022 Poster_

### Official Review · Reviewer_vB6x · 2022-10-15

**Overall Score:** 6
**Confidence:** 3

**Review:**

## Summary

In this paper, the author proposed a bi-level optimization approach that uses the RL agent to reach the desired outcome of the actions and then solves the policy as a convex problem to help achieve those states.
This proposed method can help attach problems modeled as dynamic network flow. And at the same time, it can overcome the traditional approaches, which are hard to scale up and are more efficient in solving complex dynamic network flow problems because of the nonlinearity and stochastic nature of reality.

## Pros and Cons:

### Strength:
The strong points of this paper are 1. First, it justifies their idea both theoretically and experimentally. 2. Combining the idea of combing the RL method to solve dynamic network control greatly reduces the computational complexity while the network is large-scale.
### Weakness:
The weak point of this paper is that the author stresses the advantage of this approach is it can solve dynamic network problems in a complex system, but the experiments do not justify this. It will be convincing if they can provide an experiment under a real-life scenario.

I suggest accepting this paper since using the graph reinforcement learning method to solve dynamic network control is a direct and reasonable approach that has not been fully discussed yet. And the author provides different experiments to support this though all experiments are not as complex as real-world scenarios.

One question is that since the network is dynamically changing, the action space could be changed under different current states. How does the author deal with this problem? For example, in the transversal experiment, the connectivity of the current node may vary.

---

### Official Review · Reviewer_VpxQ · 2022-10-19

**Overall Score:** 6
**Confidence:** 5

**Review:**

The paper formulates solving a dynamic network flow problem as a reinforcement learning problem where GNNs are used to represent both the policies and the value functions.

In this formulation, the (infinite) action space is the joint set of flows and exchange weights. Instead of sampling from this space, authors propose first to sample a next state s' and subsequently find an action to reach s' according to the solution of one-step Linear Control Problem (LCP). The LCP otpimization is convenient here because it can compute efficiently an action which optimizes an immediate cost as well as satisfy the system' constraints.

Overall the paper is well written and addresses an interesting problem. On the positive side, this work presents a good idea and describes reasonably well the technique. On the negative side, the proposed methodology is only illustrated on very small problems and short-horizon tasks, so the potential of using RL is not really showcased. The clarity of the paper can also be improved significantly.

The description of the learning architecture and the choice of RL method is too short. For example, regarding the GNN architecture, the presentation lacks some motivation and justification. Is the intuition that the learned representations at each layer (k) captures the relevant features at time-step (t)? Why using GNNs is suitable for representing a value function, or a policy, in this case? Also, the paper mentions A2C, which operates on a set of explicit actions, but it is unspecified how this fits into the bi-level optimization framework, e.g., how policies defined only over next states are used to encode flows and exchange weights?

I have the following additional comments/questions:

- First, the long abstract format for this paper is clearly not the best choice, since the Appendix contains fundamental information to understand the paper. Several points are confusing, e.g., the action space is defined formally only in line 383. Eqs (4a-c) refer to other Eqs. that appear later in the appendix. Consequently, the reader ends up switching back and forth between the main text and the Appendix, which contains the detailed description but refers to previous text. I suggest the authors fix this or submit a long-format paper.

- Choice of infinite-horizon instead of finite-horizon: it seems more natural to express this as a finite-horizon problem. Why should one use the discounted setting instead? What value of \gamma did you use? This needs clarification, since \gamma determines what is the temporal scale of the problem.

- Typically, policies map a state into an action (or a distribution over actions). In this work, the policy is defined over next states instead. This needs to be clarified, e.g., making explicit what is the policy at one level (a single stochastic action) and what is the policy at the other level (a deterministic policy that maps pairs of states to flows and exchange weights).

The notation: state s for the MDP are features, but s indicates commodity values. k indicates commodity index but also layer in the GNN, etc.

capapble -> cappable

---

### Official Review · Reviewer_3eyU · 2022-10-20

**Overall Score:** 6
**Confidence:** 4

**Review:**

## Summary

This work addresses network control on graphs. Specifically, it adopts a network flow model where one or multiple commodities with sources and sink nodes are routed over a graph, and nodes have the ability to exchange commodities. The goal is to control the flows and exchanges of commodities such as to maximize a monetary quantity. A typical approach is to use a linear model to make the task computationally tractable, and to use e.g. Model-Predictive Control. Instead, the authors propose an approach based on reinforcement learning, modeling the task as an MDP. The action space (arguably the most computationally challenging component of the problem) relies, instead of explicitly enumerating the decision variables, on a two-step process in which the agent's policy specifies a desired next state, and a traditional linear control method is applied. The policy is parametrized using a MPNN, and trained using a standard deep RL algorithm. The authors evaluate their framework on several network control scenarios that include dynamic topologies, constraints on the capacity of links, and distributing multiple commodities. The results show good performance compared to baselines, and reasonable performance compared to the MPC oracle, which is more expensive computationally, especially as the size of the problem increases.

## Strengths
S1. The paper addresses a family of problems of substantial interest with many high-impact applications.

S2. The formulation proposed by the authors is able to capture various realistic scenarios with promising results, and is more flexible than traditional methods.

S3. The paper is well-written, mostly clear, and to the best of my knowledge the method is sound.

## Weaknesses
W1. While the results show substantial promise, the evaluation is superficial, on small-scale synthetic scenarios. This is also acknowledged by the authors in their conclusion. Most importantly, the claim of the approach being better than MPC for e.g. nonlinear dynamics is not supported by evidence.

W2. Setting aside the merits of the paper's contributions, it is, in my opinion, too ambitious in its scope for the format of an extended abstract. It is not possible to understand the contents of the 4-page paper without checking the 9-page appendices in detail. To me, the scope of the work reads as setting a research agenda to pursue a very sensible and promising class of methods, rather than a short extended abstract.

## Comments
C1. It is not clear, in my opinion, what the decision variables are, and they should be spelled out. Lines 84-85 say "corresponds to aforementioned decision variables", but they are not mentioned as such previously. More specifically, it is not clear what are the decision variables that are used to distribute the flows -- I could not find this anywhere in the paper, but perhaps I missed it Are they "split ratios" that specify how much of a commodity the node should send to each of its neighbours along the outgoing edges? Are they something else?

C2. It is unclear to me why the MPC is labelled as "Oracle". This term suggests that the MPC knows something that the other approaches don't. Aren't the MPC and RL-based approaches all given the same knowledge of a linear model of the system? If this is the case, the use of this term is potentially misleading.

C3. Results table should present confidence intervals / error bars across multiple training runs.

C4. Appendix E (lines 547-568) mentions that GNNs are all formed of three layers. In cases where the distance between sink and source is >3 (e.g. in the four-hop case), would this not lead to some of the information not being propagated?

## Recommendation
While the paper does have weaknesses in its current form, I believe that the strengths outweigh it. I recommend acceptance and I hope the comments will help the authors reflect on their work.

---

### Official Review · Reviewer_ncLe · 2022-10-20

**Overall Score:** 5
**Confidence:** 3

**Review:**

This paper reports on an architecture containing two components for the solution of dynamic network flow modelling under particular constraints, stochasticity and complexity requirements. The goal of this architecture is to help the presented optimization problem overcome scaling issues and tractability concerns in cases where the complexity increases. The researchers present the formulations of the solution, the selection of adequate graph topologies representative of the presented problems as well as different (typically message-passing) architectures that are used in the implementation. It is clear and also written that this is a paper with the preliminary theory, framework and results of an idea that needs to be explored further.
The paper does contain significant content to justify a publication; nevertheless, the overall critique from my point of view is that it is a proposal paper, containing still a lot of heuristics and assumptions without comparing its performance with other methods directly but only through the theoretical basis – which is (according to the writers) – for some criteria superior. To have more complete scientific work and be more well-rounded those elements are crucial.

Since the main idea for the architecture starts from the acknowledgement that the dynamic network control problems are practically nonlinear, therefore they split the problem into two parts; but in lines 113-117 they already mention under which prerequisites this (approximation) is expected to work as desired. It is completely unclear though why the presented formulation (set of equations 4) is adequate to solve the problem, under which exact conditions this is not the case and what are the fundamental differences in the implementation and the results for (at first) simple problems without making it bi-level. It is not clear if they used different metrics d(.,.)in line 103 and how they, in combination with the Reinforcement Learning algorithm they’ve played a role in finding a faster/slower near-optimal trajectory.

The writers explain that one of their major motivations for this work was that previous research was based on heuristics or approximation algorithms. Nevertheless, especially in the E.2. Environments section, they do present a lot of predefined values for the used variables. Appendices A and B are very good written and the requirements in lines 369-376 are clearly stated; where it starts to get confusing is in B.1 “Dynamics” where some ideas and terms become relative (“few assumptions”, “randomly drop out”). To which degree is the “possible stochasticity” modelled? If I had a problem with the formulation presented I would not be able to know if I could/should use this method and have a profit or even if I didn’t – why. Although section C.4. and even section C.2. seem promising, we still need more directives and practical guidelines for the modelling – particularly for the stochasticity and in connection to the randomness present in E.2. Environments. How are those selected values in E.2. in accordance with and representative of specific scenarios? Even if this solution is more computationally efficient, are there any downsides?

The description of the reward is in section B.1. is too small. It is not clear how and why this reward is selected, and what is its benefits over different reward strategies. Some examples from training and test run with this and other schemes need to be made and in particular to see if the RL algorithm is capable of “sacrificing” immediate reward for the sake of long-term accumulated reward. What kind of policies are expected by this reward, how do humans characterize them? Do the expectations fit reality? Similar explanations are already provided in the “Exploration” subsection but are about the penalty terms (and still there relative using terms like “large”).

What is very characteristic of this paper is that although there is enough mention of the elements that are not favourable for other methodologies, there are no direct comparisons with them. Please specify what other output formulations – apart from the Dirichlet - you envision for future work. Table 1 contains lots of details but no interpretation comparing the results of the different architectures; each of them is regarded kind of independently.

Furthermore, even in future work, there is no reference to any Explainable AI (xAI) method that would help the agent understand if and how the model(s) base their decision-making process on solid terms and what can researchers do in case of sub-optimal or non-expected solution. When in the description of the experiments in section 4 an interpretation is argued (lines 134-136), this could be supported by xAI methods. The same goes for cases described in 149-151; some infeasible states should be detected and presented along with their characteristics.

The selection of the Dirichlet pseudocounts as being one decides for an uninformative “weak” prior. Are there thoughts of incorporating background human knowledge by making it “stronger” and more informative?

The paper is well written in general with no typos found and no unexplained variables in mathematical equations. In the “Dynamics” part of section B.1. there could be a point (2). The Outlook section does not provide for me clear directives about what future work will be.

---

### Meta-Review · Area_Chair_jNPN · 2022-11-14

**Confidence:** 3
**Recommendation:** Borderline and needs further discussi…

**Meta Review:**

# Summary

This paper proposes a graph network-based framework that can handle a wide variety of problems related to dynamic network flow models. The idea is to learn efficient solving algorithms without significantly compromising the optimality of the solution. Briefly, the problem is divided into two phases, yielding a bi-level optimization solving process. Experiments are carried out on the minimum cost flow problem and the results show good performance compared to baselines, and reasonable performances compared to the oracle, which is more expensive computationally.

# Recommendation

This paper was well received by the reviewing team and the author response was helpful to converge towards this agreement. The concerns and questions were successfully addressed by the authors. Here are the three main points motivating this recommendation:

1. The framework proposed is generic and flexible, in the sense that it can handle different dynamic network control problems. However, more experiments should be carried out to confirm this point.

2. The problem targeted is interesting, and efficient solving procedures may have an high-impact in real applications.

3. Although the paper is well-written, many critical information are only available in appendices (formalization of the mathematical model, RL environment, GNN architecture), which makes the paper not self-contained for an extended abstract. It is not possible to grasp the paper and seize the impact of the experiments without back-and-forth with the appendices.

# Conclusion

To conclude, the reviewing team think that the paper has merits to be accepted. However, I would like to highlight that I do not think that this kind of paper, with a lot of critical content in appendices, is the most suited for an extended abstract. This consideration is also reflected by Reviewer 3eyU.

---

### Decision · Program_Chairs · 2022-11-23

Accept (Poster)